# From space to time: Spatial inhomogeneities lead to the emergence of spatiotemporal sequences in spiking neuronal networks

Sebastian Spreizer[1,2], Ad Aertsen[1,2], Arvind Kumar[2,3]*

**1** Faculty of Biology, University of Freiburg, Freiburg, Germany, **2** Bernstein Center Freiburg, University of Freiburg, Freiburg, Germany, **3** Computational Science and Technology, School of Electrical Engineering and Computer Science, KTH Royal Institute of Technology, Stockholm, Sweden

\* arvkumar@kth.se

## Abstract

Spatio-temporal sequences of neuronal activity are observed in many brain regions in a variety of tasks and are thought to form the basis of meaningful behavior. However, mechanisms by which a neuronal network can generate spatio-temporal activity sequences have remained obscure. Existing models are biologically untenable because they either require manual embedding of a feedforward network within a random network or supervised learning to train the connectivity of a network to generate sequences. Here, we propose a biologically plausible, generative rule to create spatio-temporal activity sequences in a network of spiking neurons with distance-dependent connectivity. We show that the emergence of spatio-temporal activity sequences requires: (1) individual neurons preferentially project a small fraction of their axons in a specific direction, and (2) the preferential projection direction of neighboring neurons is similar. Thus, an anisotropic but correlated connectivity of neuron groups suffices to generate spatio-temporal activity sequences in an otherwise random neuronal network model.

## Author summary

Here we propose a biologically plausible mechanism to generate temporal sequences of neuronal activity in network of spiking neurons. We show that neuronal networks exhibit temporal sequences of activity when (1) neurons do not connect in all directions with equal probability (asymmetry), and (2) neighboring neurons have similar connection preference (spatial correlations). This mechanism precludes supervised learning or manual wiring to generate network connectivity to produce temporal sequences. Connection asymmetry is consistent with the experimental findings that axonal and dendritic arbors are spatially asymmetric. We predict that networks exhibiting temporal sequences of neuronal activity should have spatially asymmetric but correlated connectivity. Finally, we argue how neuromodulators can play a role in rapid switching among different temporal sequences.

**Data Availability Statement:** The simulation code is available at https://github.com/babsey/spatio-temporal-activity-sequence.

**Funding:** The work was funded in parts by the German-Israeli Foundation (G-1222-377.13/2012), Carl-Zeiss Foundation, Vetenskapsrådet (StratNeuro, India Sweden Grant, Research Project Grant), and Parkinsonfonden. The funders played no role in study design, data collection and analysis, decision to publish or preparation of the manuscript.

**Competing interests:** The authors have declared that no competing interests exist.

# Introduction

Ordered sequences of actions are the key to any meaningful behavior. This implies that the task-related neuronal spiking activity in the task-related brain regions must also be ordered in temporal activity sequences [1, 2]. Indeed, temporal activity sequences have been recorded from different brain regions in various tasks [3–11] (see [12] for a review). The necessity and ubiquity of sequential activity patterns in the brain raises the question: What is the origin of such activity sequences in locally random, sparsely connected networks of noisy neurons?

At the simplest, activity sequences of neurons may be attributed to their external inputs. When neurons are tuned to specific properties of an external input, a sequential change in the input can lead to an activity sequence, e.g. temporally ordered spiking of place cells in the hippocampus [13]. However, activity sequences have been observed in tasks that do not involve any specific sequential stimuli, e.g. in decision making [7, 8], in learning [10], in memory recall [6], and in generating bird songs [3]. This suggests that neuronal networks in the brain are able to generate neuronal activity sequences using intrinsic mechanisms.

A feedforward network [14] is the simplest model that can generate activity sequences [15–17]. However, given the random and recurrent connectivity in the brain, this architecture is biologically disputed. Recurrent network models with an asymmetric spatial connectivity can exhibit traveling waves [18–21], which can be considered as a spatio-temporal activity sequence. However, in this dynamical regime a network essentially generates a single activity sequence. Recurrent networks tuned to exhibit attractor dynamics [22] can generate more diverse temporal activity sequences in response to an external input which steers the spiking activity across attractor states [23]. Alternatively, spike-frequency adaptation, spike threshold adaptation and short-term synaptic depression could also underlie the emergence of activity sequences [24–26], although switching from one attractor to another is stochastic and depends on the noise level and initial conditions. Reliable sequential switching between attractors often requires manual wiring of neurons representing different attractors in a feedforward manner [26]. Beyond attractor networks, more generic echo-state-networks can be trained using a supervised learning algorithm to generate an arbitrary temporal sequence of neuronal activity [27]. Recently, biological synaptic timing-dependent plasticity rules have also been used to modify the connectivity of recurrent networks to generate activity sequences [28]. Overall, previous research suggests that the emergence of activity sequences in a recurrent neuronal network model requires one or more of the following features: external inputs, spike frequency adaption, synaptic depression and (supervised or unsupervised) learning. These proposals for sequence generation implicitly assume that innately the networks are 'tabula-rasa' and emergent network dynamics or learning create sequential activity.

Here, we present a connectivity rule that can endow a recurrent neuronal network with the ability to generate reliable sequential activity (spontaneous as well as evoked) without explicitly relying on learning or emergence attractor dynamics. We studied the emergence of the activity sequences in a neuronal network model with distance-dependent connectivity. We show that when the extent of the spatial connectivity is asymmetric and varying across neurons, spatio-temporal activity sequences (STAS) emerge. We have identified two conditions that ensure the emergence of STAS in a network with spiking neurons: (1) individual neurons project a small fraction ($\approx$2-5%) of their axons in a preferred direction $\phi$. (2) $\phi$s for neighboring neurons are similar, whereas $\phi$s for neurons further apart are unrelated. These conditions do not depend on the composition of neurons in the network and both, purely inhibitory networks (e.g. networks in the striatum and central amygdala) and networks with both excitatory and inhibitory neurons (e.g. those seen throughout the neocortex) can exhibit STAS, provided the two conditions mentioned above are met. Thus, we present a generative connectivity rule resulting in a

network model that can exhibit STAS in its spontaneous as well as evoked activity states. Being a connectivity rule, it essentially hardwires STAS into the network. To alleviate this restriction we propose a mechanism by which neuromodulators can modify the network connectivity dynamically to generate and modulate STAS at behaviorally relevant time scales.

## Results

Sequential activity requires that there are feedforward networks embedded in an otherwise recurrent random network. Here we investigate if there exists a general connectivity rule that can create locally connected random networks (LCRN) with feedforward networks embedded in them, without explicitly embedding feedforward subnetworks [17] or learning them [27]. In other words, can a recurrent network be 'innately' wired to generate STAS. It is well known that LCRNs can exhibit stable hexagonal patterns of activity bumps [21, 29, 30]. We hypothesize that such stable spatial activity patterns can be transformed into STAS if the activity bumps could be destabilized. To this end, we investigated the effect of introducing inhomogeneities in the spatial connectivity between neurons on the stability of the activity bumps.

### Spatial distribution of inhomogeneities in neuronal connectivity

We considered an LCRN in which neurons projected a fraction of their axons preferentially in a particular direction ($\phi$; Fig 1a and S1b Fig). $\phi$ was chosen from a uniform distribution and assigned to each neuron according to four different configurations (Fig 1b). Random configuration: $\phi$ was randomly and independently assigned to each neuron. Perlin configuration: $\phi$ was assigned to neurons using a gradient noise algorithm such that neighboring neurons had similar values of $\phi$. Homogeneous configuration: the same $\phi$ was assigned to all neurons. Finally, as a control, we also considered the case in which all neurons projected in all directions with equal probability (Symmetric configuration).

First, we focused on LCRNs with only inhibitory neurons (I-networks). In these I-networks, we used a connectivity profile which varied non-monotonically with distance, according to a Gamma distribution (Fig 1a:center; see Methods [30]). After wiring the networks according to each of the four configurations described above, we measured the effective $\phi$ from the spatial distribution of the post-synaptic targets of each neuron (Fig 2a1–2a5). For the random and Perlin configurations, the angle $\phi$ measured from the location of the post-synaptic neurons was uniformly distributed, as was initially specified. For the homogeneous configuration all neurons had identical $\phi$ assigned, but the measured $\phi$ values for individual neurons were slightly different from the assigned value, due to the finite numbers of connections per neuron. For the whole network, $\phi$ was normally distributed around the assigned value, with a very small variance. In the symmetric configuration, $\phi$ for the network was uniformly distributed and was different for each neuron, due to the random nature of the connectivity and the finite numbers of connections per neuron.

The in-degree distribution was similar across all four configurations (Fig 2b1–2b5). However, in the Perlin configuration, as a consequence of the spatial distribution of $\phi$, neurons with high and low in-degree distribution were spatially clustered. Because the connectivity of the recurrent inhibition was spatially symmetric in EI-network models, the spatial distribution of the excitatory in-degree reflects the spatial distribution of the excitation and inhibition balance (EI-balance). This landscape of the EI-balance, of course, will be modulated by the network activity state. Overall, the networks were highly similar across all four configurations at the level of neuron properties and their connectivities (same in-degree distribution and fixed out-degree for all neurons).

## Schematics of the asymmetric networks

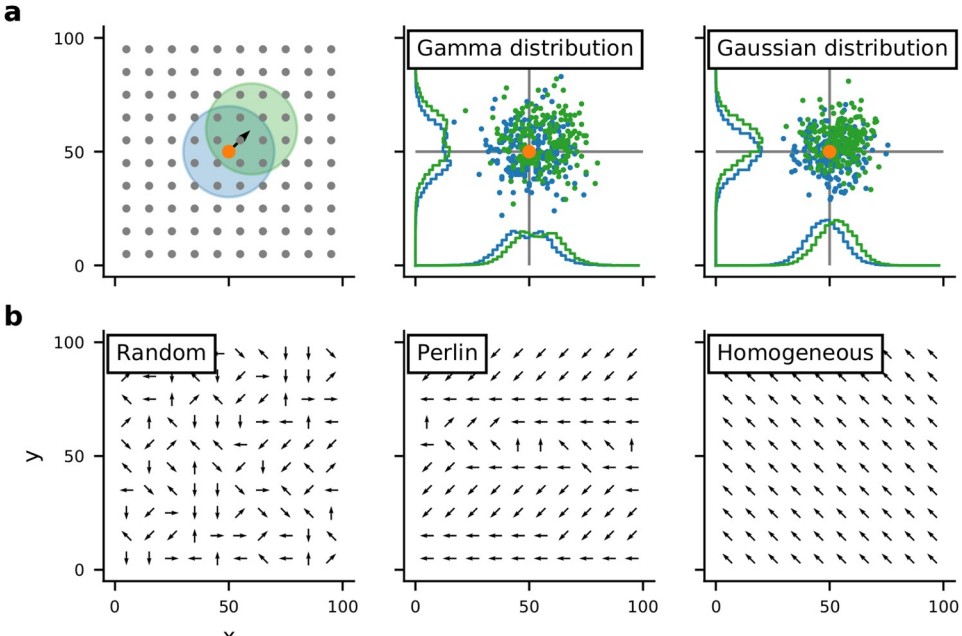

**Fig 1. Schematics of the asymmetric network models. (a:left)** Neurons were arranged on a regular 2-D grid, folded to form a torus. The colored circles indicate the symmetric (blue) and asymmetric (green) spatial connectivity schemes. The pre-synaptic neuron is marked by the orange dot. **(a:center)** Locations of post-synaptic neurons chosen according to the asymmetric (green) or symmetric (blue) connectivity. In this case the distance-dependent connectivity profile varied non-monotonically, according to a $\Gamma$ distribution. This connectivity profile was used for purely inhibitory network models. **(a:right)** Same as in the center panel, but here the distance-dependent connectivity profile varied monotonically according to a Gaussian distribution. This connectivity profile was used in the present study for network models with both excitatory and inhibitory neurons. **(b)** Schematic of spatial distribution of connection asymmetries. Each arrow shows the direction in which the neuron makes preferentially most connections ($\phi$). Here we show examples for random, Perlin and homogeneous configurations.

## Spatial inhomogeneities lead to the emergence of activity sequences

The differences among the four connectivity configurations became evident as we inspected the corresponding network activity dynamics. To generate network activity we injected independent Gaussian white noise into each neuron (see Methods). In an LCRN with Perlin configuration, time-resolved snapshots of the activity showed transient co-activation of neighboring neurons, referred to as spatial activity bumps (see S1 Video). Importantly, the spatial bumps were not fixed at a given location, instead as one spatial bump faded, another, similar bump appeared in its immediate vicinity, and so on, thereby creating STAS. Because we did not implement short-term synaptic depression or spike frequency adaptation, the silencing of a spatial bump was a consequence of the network's dynamical activity state and of the spatial $\phi$-distribution. Time-averaged firing rates (estimated over 10 sec) showed that neurons participating in the activity sequences were arranged in stripe-like patterns in the network space (Fig 2c), along which the activity sequences flowed.

We used the DBSCAN algorithm (see Methods) to track spatial bumps of spiking activity over time to identify the activity sequences (Fig 3a). The identified STAS followed specific paths in the network, visible as stripes in the spatial distribution of average firing rates of individual neurons. Each sequence moved in its own direction, the collection of them forming a uniform distribution of activity sequence movement directions (Fig 2d5).

## Measurements of the spiking activity in I networks

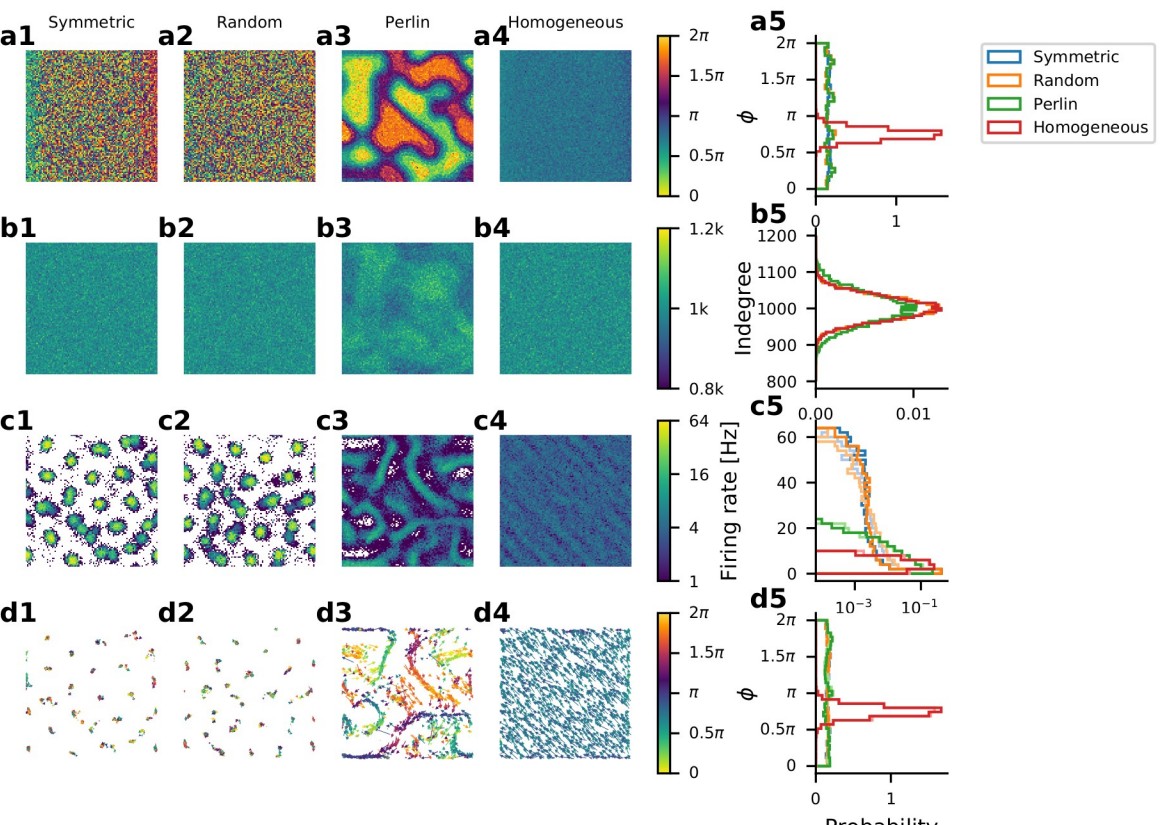

**Fig 2. Network structure and spiking activity in I-networks. (a)** Spatial distribution of connection asymmetries. The square represents the 2-D space of the network. The four panels (**a1-a4**) show the four different configurations of asymmetric connectivity: symmetric, random, Perlin and homogeneous. The panel **a5** shows the distribution of $\phi$, measured for each neuron from the locations of its post-synaptic neurons. **(b)** Spatial distribution of in-degrees of individual neurons in the four configurations (**b1-b4**). The in-degree distribution was similar for all four configurations (**b5**). Note that in the Perlin configuration, neurons with high and low in-degree were spatially clustered (**b3**). **(c)** Spatial distribution of average firing rates of individual neurons in the four network configurations (**c1-c4**). **(c5)**The distribution of firing rate of all the neurons. **(d1-d4)** Spatially distributed direction of neuronal activity flow in the four configurations. **(d5)** Distribution of the direction of neuronal activity flow independent of space. In symmetric, random and Perlin configurations, activity could move in all possible directions (blue, orange, green), whereas in the homogeneous configuration, activity flowed in a single direction (red). Note that in symmetric and random configurations, despite the presence of all possible directions of projection, the network activity remained locked at certain specific locations (**d1,d2**), unlike in the Perlin configuration, in which a clear and spatially diverse flow of activity emerged (**d3**).

In the homogeneous configuration, an extreme case of the connectivity asymmetry, the network activity exhibited multiple moving bumps. Neurons participating in moving activity bumps were arranged in a periodic pattern (Fig 2c4) and the activity sequences flowed along the associated stripes (Fig 2d4). Such patterns of average activity flow closely resembled the 'static' patterns observed in bio-chemical systems [31]. Unlike in the Perlin configuration, in the homogeneous configuration all spatial bumps moved in the same direction (Fig 2d5, red trace). Because knowing the movement direction of a single activity sequence was sufficient for knowing the movement directions of all other sequences, the homogeneous configuration effectively exhibited only a single spatio-temporal activity sequence. This type of activity pattern was similar to the traveling waves observed in neural field models [21, 29].

## Sequential activity of the Perlin networks

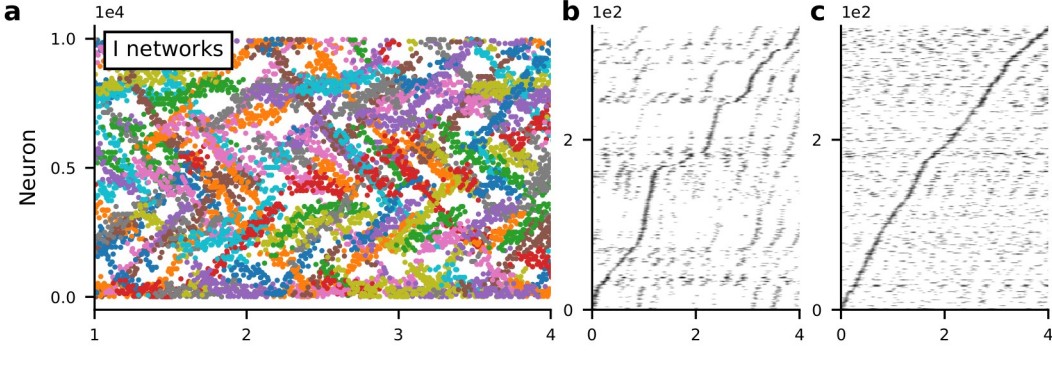

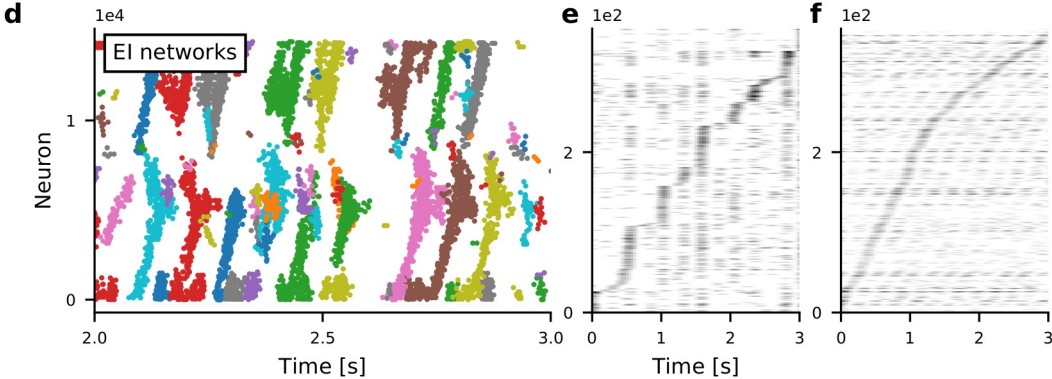

**Fig 3. Spatio-temporal sequences of neuronal activity in networks with Perlin configuration. (a)** Raster plot of spiking activity in an I-network model as a function of time. Each color indicates a cluster of spiking activity in space and time, identified using DBSCAN (see Methods). Note that spikes that were not assigned to any cluster are not shown. **(b)** Activity of approx. 250 neurons confined in a $20 \times 20$ region. **(c)** Activity of approx. 250 neurons randomly selected from the entire network. **(b, c)** Selected neurons are sorted according to the time of peaked spike counts. **(d, e, f)** Same as in panels **a, b, c**, respectively, but here for an EI-network model. Note the shorter time axes in panels **d, e, f**, compared to panels **a, b, c**, indicating that sequence movement in EI-networks was clearly faster than in I-networks.

When $\phi$ was distributed randomly (random configuration) or when neurons made connections without any directional bias (symmetric configuration), we did not observe any STAS. In both configurations, the network activity was confined to specific neurons, while others were inhibited, giving rise to a long-tailed distribution of average firing rates (Fig 2c5). In both symmetric and random configurations, active neurons were organized in a near hexagonal pattern of spatial activity bumps (Fig 2c1 and 2c2). Such an activity pattern is a consequence of the non-monotonic shape of the effective connectivity [30]. In the random configuration, the spatial organization of the activity bumps was a bit more noisy than in the symmetric configuration. In both configurations, the spatial bumps jittered randomly around a fixed location, resulting in a uniform distribution of bump movement directions (Fig 2d1, 2d2 an 2d5). Thus, both random and symmetric configurations resulted in similar types of network activity states.

Similarly to the I-network models, an LCRN with both excitatory and inhibitory neurons (EI-network) also exhibited STAS when excitatory neurons made connections to excitatory neurons preferentially in one direction and $\phi$-values were distributed according to the Perlin configuration (Fig 3d). In both EI- and I-network models, the activity sequence could be extracted from only a few neurons chosen from a small neighborhood (Fig 3b and 3e) or

randomly from the whole network (Fig 3c and 3f). When the spiking activity was sampled from the entire network and neurons were ordered according to their peak firing rates (as is often done with experimental data [8, 11]), the velocity of the activity sequence appeared to be quite constant (see Fig 3c and 3f). Experimental data suggest that the velocity of temporal sequences can vary over time [11]. In our network model, we also found that when about 250 active neurons were sampled randomly from a small network neighborhood, the velocity of the activity sequences varied as a function of time (Fig 3b). However, this varying velocity could be an artefact of the finite size effect and of the non-uniform sampling of the sequences (see Fig 3b and 3c). In general, the activity sequences in EI-network models were faster than those in I-networks, because the activity sequences in EI-networks relied on recurrent excitation, whereas in I-networks they relied on the lack of recurrent inhibition (in our I-networks, the neuronal connectivity varied non-monotonically with distance, according to a Gamma distribution and, hence, had only a small connection probability among neighboring neurons).

### Conditions for the emergence of spatio-temporal activity sequences

These results suggested that the emergence of STAS in LCRNs required two conditions to be met: (1) each neuron projects a small fraction of its axons preferentially in a specific direction ($\phi$), and (2) neighboring neurons preferentially project in similar directions, whereas the projection directions of neurons further apart are unrelated. These two conditions imply a spatially correlated anisotropy in the projection patterns of neurons in the network. Indeed, upon systematic variation of a wide range of input parameters and excitation-inhibition balance, we found that, as long as these two conditions were met, irrespective of the composition of neurons in the LCRN, STAS invariably emerged (S2 Fig).

### Co-existence of activity sequences and network oscillations

The rasters of spiking activity in both I-networks and EI-networks indicated the presence of slow oscillations in Perlin (Fig 3) and homogeneous configurations. Therefore, we measured the spectrum of the summed network activity. This summed network activity was obtained by different procedures: by summing the activity of all neurons (Fig 4, blue trace), by summing the activity of the neurons from a 10×10 region in the network (Fig 4, green trace), and by summing the activity of 100 randomly chosen neurons from the entire network (Fig 4, orange trace). For I-networks (Fig 4a), neuronal population activity in all four configurations exhibited clear oscillations in the gamma frequency band ($\approx 60$ Hz). These oscillations were a global property of the network, and partial sampling of the neurons also showed oscillatory activity, albeit with lower power. (Fig 4a, orange and green traces). Moreover, in homogeneous and Perlin configurations, signs of low-frequency oscillations at around 3-4 Hz in I-networks were observable. These were presumably a consequence of the periodic boundary conditions, i.e. the period of slow oscillations was determined by the sequence propagation velocity (see below) and the spatial network size. Oscillations in the EI-networks were observed at $\approx 30$ Hz, however, the oscillation power was smaller than in the I-networks. A closer inspection of oscillations in the EI-networks revealed that oscillations occurred in short bursts, resulting in smaller oscillation power (S3 Fig). These results suggest that both STAS and global oscillations can co-exist both in I-networks and in EI-networks.

### Asymmetry in connectivity determines the velocity of STAS

Next we investigated how the amount of shift in the connectivity affects the STAS. To this end we shifted the connectivity extent by 1 and 2 grid points. Shift by one grid point increases connectivity in the direction $\phi$ by 2-5%, depending on the distance and a corresponding decrease

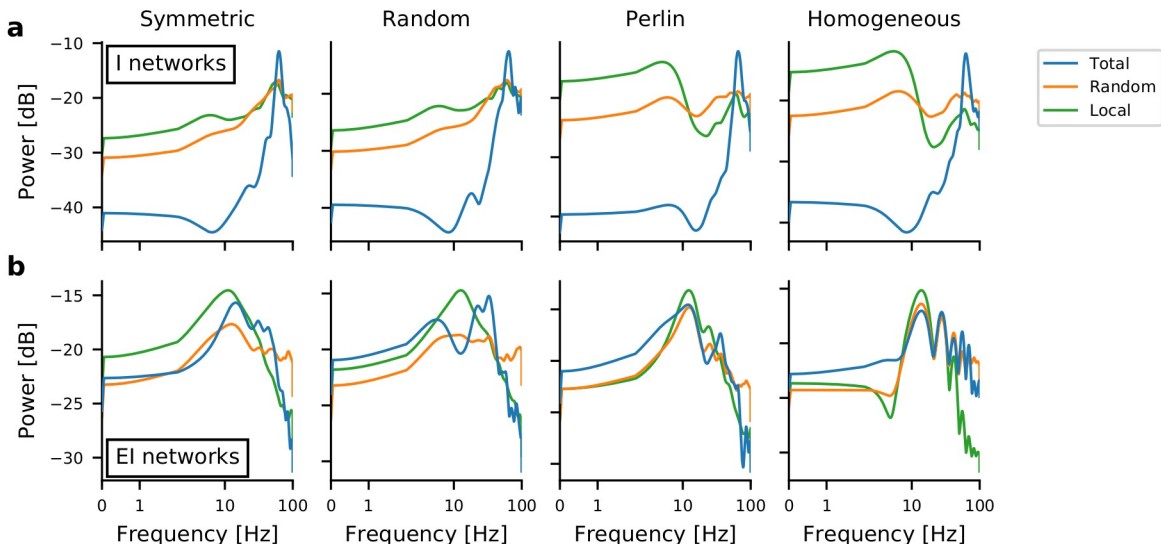

**Fig 4. Power spectra of network activity in I-networks and EI-networks in different spatial inhomogeneity configurations.** (**a**) Power spectra of summed spiking activities (bin width = 5 ms), with different traces referring to the source of the data: the z-score of the spiking activity of the entire network population (blue trace), of 100 randomly selected neurons from the entire network (orange trace), and of the neurons located in a 10×10 region in the network (green trace). The power spectrum in all I-network models peaked at approx. 60 Hz (gamma-band oscillations). Additionally, in network models with homogeneous and Perlin configurations, an additional, weak low-frequency peak, at around 3-4 Hz, appeared. (**b**) Same as in **a** for EI-networks. Low-frequency peak around 12-14 Hz were observed in networks with homogeneous and Perlin configurations.

in the direction opposite to $\phi$ (S1b Fig). When there was no shift in the connectivity, networks did not exhibit any sequential activity and the activity bumps jittered around a fixed value with a small velocity (Fig 5a and 5b blue). However, shifting the connectivity by 1 grid point was sufficient to induce sequential activity in both homogeneous and Perlin configurations. The velocity of STAS was higher in the homogeneous configuration than in the Perlin configuration (Fig 5a and 5b orange). When we increased the shift in connectivity to 2 grid points (i.e. 5-10% increase in the connectivity, depending on the distance), the mean and variance of the velocity increased in both Perlin and homogeneous configurations (Fig 5a and 5b green). These results suggest that the degree of asymmetry in the connectivity controls the velocity of STAS.

### Effect of spatial correlation in connection asymmetry on spatio-temporal activity sequences

Next, we determined how the spatial correlations in $\phi$ affect the number and velocity of STAS. To this end we systematically varied the spatial scale of the Perlin noise (see Methods). This enabled us to systematically move from a random configuration to a homogeneous configuration (Fig 6a top). To count the number of STAS, we rendered the activity in a 3-dimensional space (two space dimensions and one time dimension) and used the DBSCAN algorithm to identify clusters (which are the STAS) in this 3-D space. We found that the number of STAS and their velocity decreased monotonically as we reduced the Perlin scale (Fig 6b and 6c). This decrease in number and velocity of STAS occurred because a reduction in Perlin scale reduced the number of neighboring neurons with similar $\phi$. This, in turn, reduced the velocity of the

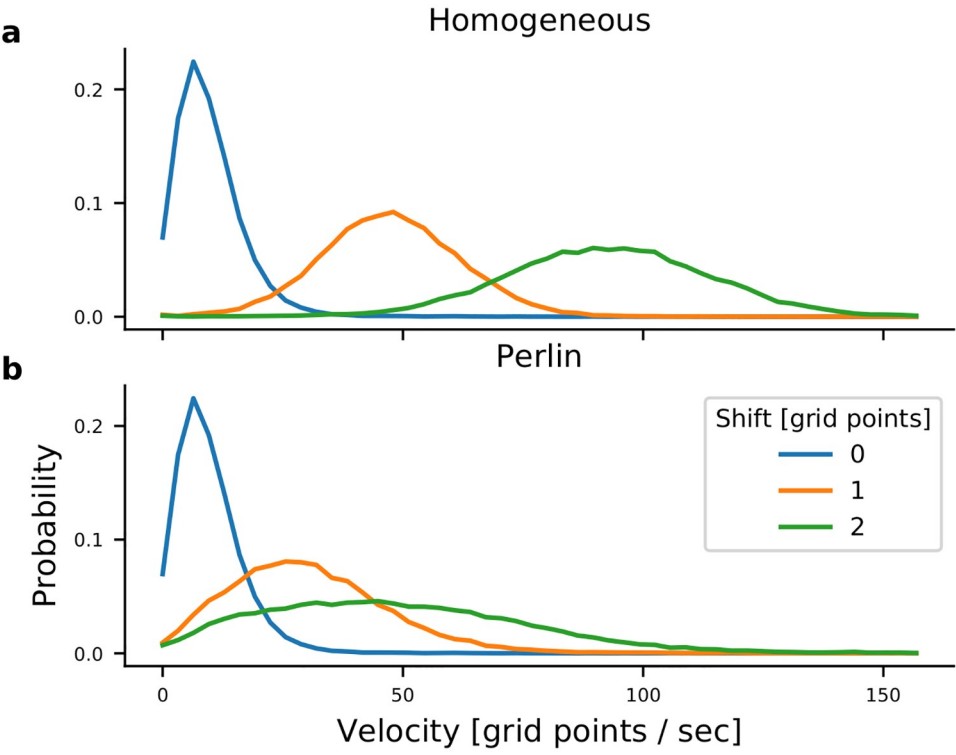

**Fig 5. Velocity of neuronal activity sequences.** Distribution of the velocity of neuronal activity sequences in I-networks with homogeneous (**a**) and Perlin (**b**) configurations. The velocity of the activity sequences increased with the increase in connectivity asymmetry. Note that the velocity of activity bump movements in networks with symmetric connections (blue traces) were identical in networks with homogeneous and Perlin configurations. However, for any non-zero degree of asymmetry (orange and green traces) the velocity of activity bump movements was higher in networks with the homogeneous configuration.

STAS (Fig 6b), because the input in the direction specified by $\phi$ decreased and the postsynaptic neurons had to integrate over longer time to elicit response spikes. Moreover, because of the fewer inputs in the direction $\phi$, many putative sequences showed weak spiking activity which could not be classified as a distinct spatio-temporal sequence. Furthermore, a reduction in Perlin scale also increased the variance of movement directions (Fig 6c). These results show, first, that even a small scale spatial correlation in $\phi$ suffices to induce STAS but, second, if the spatial correlation scale is too small, such sequences may not move quickly enough to be detected as sequences. For functionally relevant STAS, the spatial correlation (i.e. Perlin scale) in $\phi$ should be about 20, which is about $\frac{1}{6}$th of the network size.

## Origin of STAS in networks with spatially correlated connection asymmetry

To get more insight into the mechanisms underlying the emergence of STAS in Perlin and homogeneous networks we estimated the eigenvalue spectrum of the network's connectivity matrix. For an Erdós-Renyi type random network, eigenvalues of the connectivity matrix are distributed in a circle [32]. In an inhibition dominated network, extra inhibition introduces very large negative eigenvalues that contribute to the stability of the network activity dynamics [33]. Here, we found that for an LCRN without any directional connectivity (symmetric configuration), most eigenvalues were confined within a circle, but the local nature of the

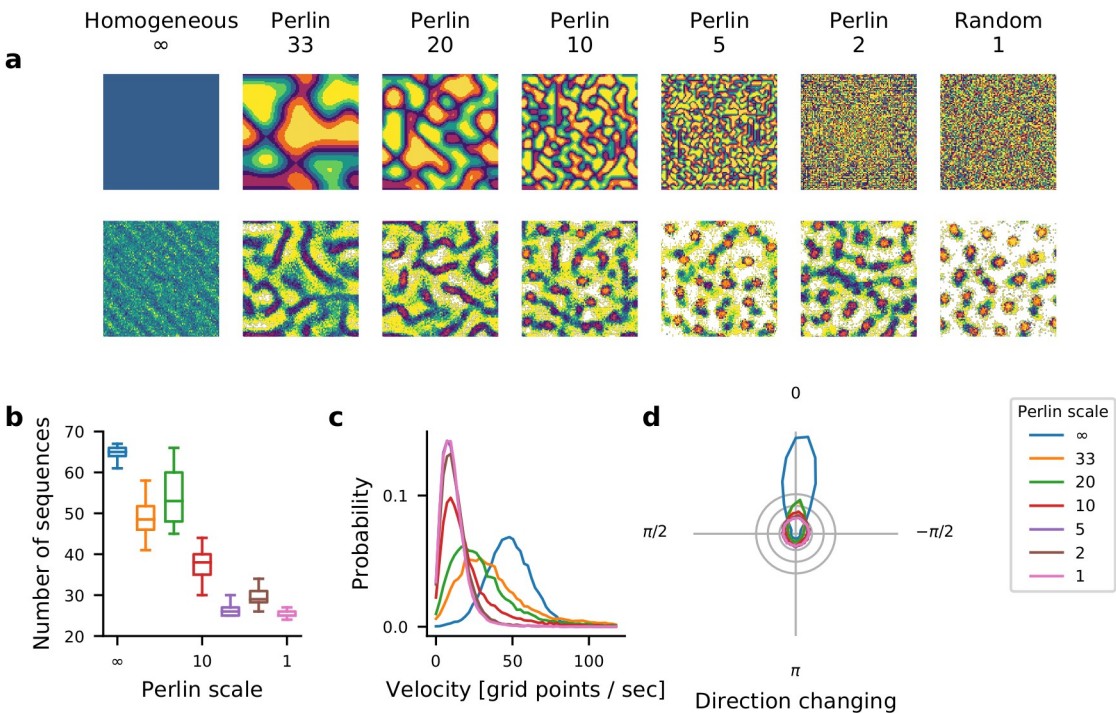

**Fig 6. Effect of spatial scale of correlations in _φ_ on the emergence and velocity of STAS in I-networks. (a)** The top row shows the spatial distribution of _φ_ for different scales of Perlin noise. The Perlin scales decreased from left to right as reflected in the size of single color blobs. The Perlin scale is indicated in terms of grid points in the network. The bottom row shows the spatial distribution of average firing rates in each of the seven configurations. **(b)** The number of STAS observed in 1 sec. for different Perlin scales. The box plot shows that statistics of STAS estimated over 90 epochs of 1 sec. each. Different colors indicate the scale of the corresponding Perlin noise. **(c)** The distribution of the velocity of STAS. Different colors indicate the scale of the corresponding Perlin noise. **(d)** The distribution of STAS directions in polar plots. In a homogeneous configuration, most sequences moved in a single direction (blue curve). As the Perlin scale decreased, the distribution of movement direction became more widely distributed, indicating an increase in the number of sequences that moved in different directions.

connectivity introduced several eigenvalues outside the circle, with large real parts and small imaginary parts. As we introduced spatial asymmetry into the connectivity, the imaginary component of large eigenvalues (those outside the circle) increased (Fig 7a). The emergence of large complex eigenvalues outside the main circle is indicative of meta-stable dynamics [32]. Moreover, the number of large eigenvalues outside the main lobe (circle in this case) of the eigenvalue spectrum is equal to the number of 'communities' of neurons in a network [34]. This suggests that in both Perlin and homogeneous configurations correlations in the spatial distribution of _φ_s created many communities (neuronal assemblies), the dynamics of which are meta-stable.

Given the large size of our network models, it is computationally highly demanding to test this hypothesis by measuring all eigenvalues of an LCRN, identifying and counting the neuronal assemblies and determining the effective feedforward networks associated with their STAS. To simplify the problem, we estimated the probability of finding such a feedforward network _pFF_ in our I- and EI-network models. To this end, we used an iterative procedure to find feedforward networks in our network models, the details of which are described in the Methods section. Briefly, we started with a set of 64 neurons ($F_i$) located in a small, 8×8 region in the network. Then we identified the set of all post-synaptic neurons ($P_i$) receiving input (excitatory input in an EI-network and inhibitory input in an I-network) from any of the neurons in the first set $F_i$. From the set $P_i$ we selected the 64 neurons ($F_{i+1}$) that were most frequently

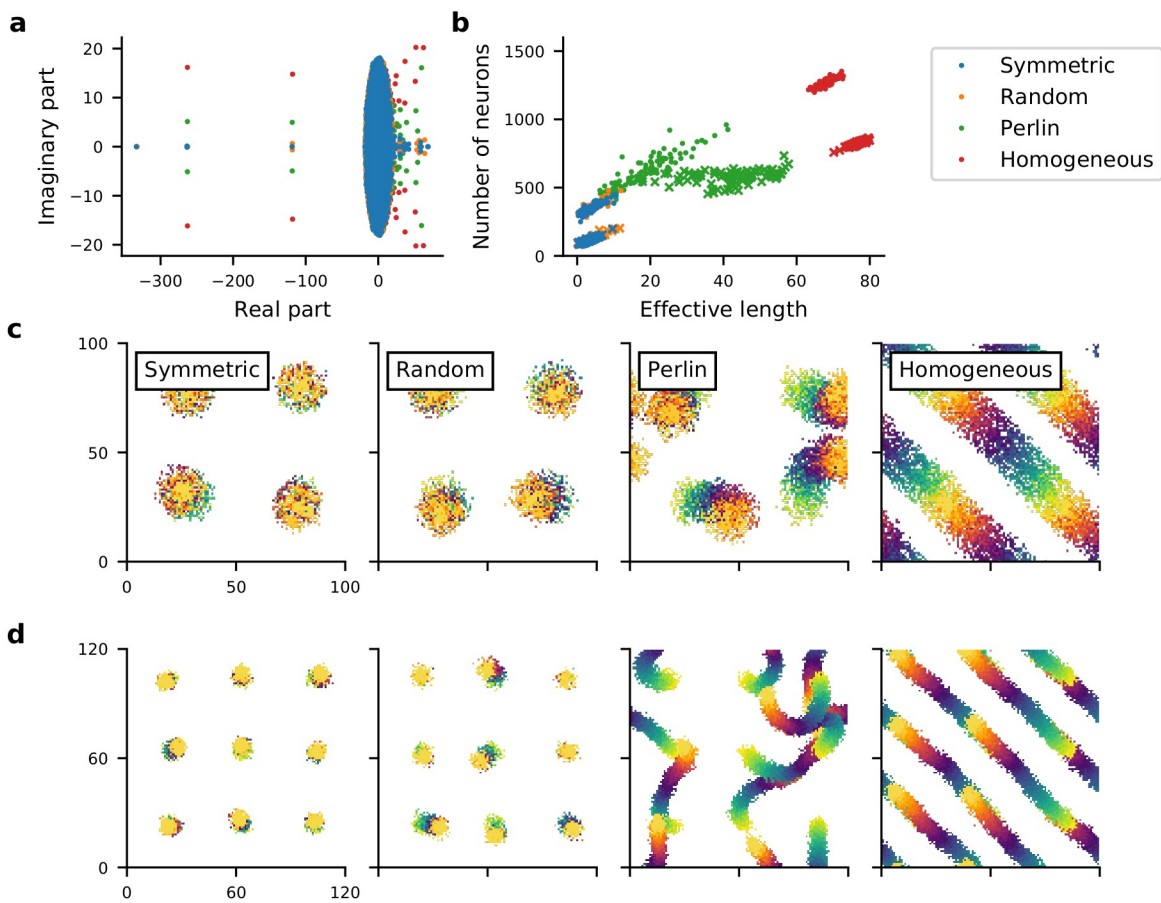

**Fig 7. Spatial clustering of $\phi$ results in feedforward pathways in otherwise locally connected random networks. (a)** The eigenvalue spectrum of the connectivity matrix of 1,000 inhibitory neurons randomly selected from symmetric (blue dots), random (orange dots), Perlin (green dots) and homogeneous (red dots) I-networks. **(b)** Number of unique target neurons participating in a feedforward path (y-axis) as a function of the effective length of the feedforward path (Euclidean distance between the centroids of $F_1$ and $F_{50}$ (see Methods). Feedforward path in I-network (dots), feedforward path in EI-network (crosses). The four colors indicate the network configurations. Note that distinctly more unique neurons with longer path length of the sequential activity movement were observed in Perlin and homogeneous configurations. **(c)** Effective feedforward pathways in an I-network model with the four configurations (see Methods). Feedforward paths starting from four different locations are shown. The starting neuron set $F_1$ is shown in yellow, the final set $F_{50}$ is shown in orange. Effective feedforward pathways were visible as trails changing color from yellow to orange. The starting neuron set $F_1$ consisted of 64 neurons located in an 8×8 region of the network. **(d)** Same as in **c**, but for an EI-network model with nine different starting set locations.

connected to the neurons in the input $F_i$ (see Methods for details). We repeated this procedure 50 times, starting at 100 randomly selected different locations of the initial 8×8 regions (see Methods, Fig 7b and 7c). Thus, we identified feedforward networks with excitatory connections from $F_n$ to $F_{n+1}$ in EI-networks and feedforward networks with inhibitory connections from $F_n$ to $F_{n+1}$ in I-networks.

In the homogeneous configuration we always found a feedforward path capable of creating a STAS (Effective length > 16; see Methods and Fig 7c and 7d). Indeed, in the homogeneous configuration, the probability of finding a feedforward path: *pFF* was 1.0 (for both EI- and I-networks). Moreover, these feedforward paths were very long (Fig 7b, red dots and crosses). In the Perlin configuration, there were fewer (*pFF* ≈ 0.8 for EI-networks; *pFF* ≈ 0.66 for I-networks) and shorter (Fig 7b, green dots and crosses) feedforward paths, but they pointed in different directions (Fig 7b–7d). By contrast, in both symmetric and random configurations, no

feedforward pathways were observed ($pFF = 0$ for both EI- and I-networks). In these latter two configurations, neurons participating in $F_1$ to $F_{50}$ were confined to a small space (indicated by the overlap of the color blobs in Fig 7c and 7d). Ultimately, it was the existence (or non-existence) of these feedforward pathways that determined the properties of STAS in the four different configurations.

Within a feedforward path in an EI-network, excitatory neurons in $F_n$ projected to excitatory neurons in $F_{n+1}$ with higher probability than outside $F_{n+1}$, thereby creating a path of high excitation between successive groups in the path. When an external input was strong enough, a STAS was observed along such path of high excitation. By contrast, within a feedforward path in an I-network, inhibitory neurons in $F_n$ projected to inhibitory neurons in $F_{n+1}$ with higher probability than outside $F_{n+1}$, thereby creating a path of high inhibition from $F_1$ to $F_{50}$. Because the out-degree of neurons was fixed, the concentration of inhibitory connections within the path from $F_1$ to $F_{50}$ created a region of low recurrent inhibition in the vicinity of the path, along which inhibitory STAS emerged. Thus, the abundance of feedforward paths in networks with Perlin and homogeneous configurations provided a structural substrate for the emergence of the rich repertoire of STAS in such networks. Moreover, through this analysis we also demonstrated that correlation in the in-degree (i.e. shared inputs) and in the out-degree (i.e. shared outputs) underlie the emergence of sequential spiking in neurons and that spatially correlated anisotropic connectivity is a simple generative mechanism to achieve such in- and out-degree correlations.

## Stimulus-evoked spatio-temporal activity sequences

In the above we described how STAS emerged in networks with spatially correlated asymmetric connectivity of neurons. Next, we investigated whether these networks could also generate STAS in response to an external stimulus. To this end, we excited a small set of 50 neighboring neurons for 50 ms and measured the network response. As expected, the network response depended on the background activity of the network (Fig 8a and 8b). Therefore, we measured the properties of the evoked STAS in different states of ongoing activity—from a silent network to a network with high background activity and spontaneous STASs (Fig 8c, S3 Fig). To obtain different states of ongoing activity we varied the input mean and variance (Fig 8c, S3 Fig).

We found that when the background activity was low (weak input regime), the network a STAS was evoked with high probability (Fig 8d). Close to the threshold at which STATS emerged in the background activity, the network response time was smallest, whereas in a completely silent network, the network response to an incoming stimulus was rather slow (Fig 8e). Finally, in the weak input regime, the evoked STAS persisted for only few milliseconds after the input stopped (Fig 8f). When there were STAS in the background activity (strong input-regime), the probability to evoke a STAS decreased, because the neurons were already engaged in the existing STASs (Fig 8d). In the strong input regime, neurons that were not a part of an ongoing STAS were strongly inhibited, therefore, the stimulus response was also slow in this regime. The long persistence of evoked STAS in the strong input regime (Fig 8e) was because the evoked STAS coincided with an ongoing STAS, giving the (false) impression of long persistence.

## Neuromodulators can generate spatially correlated inhomogenenties

Our proposed sequence generation mechanism demands another mechanism which enables a group of neurons to make more or stronger synapses in a common direction $\phi$ in the first place. Axonal and dendritic arbors of neurons are almost never symmetrical in space [35] and

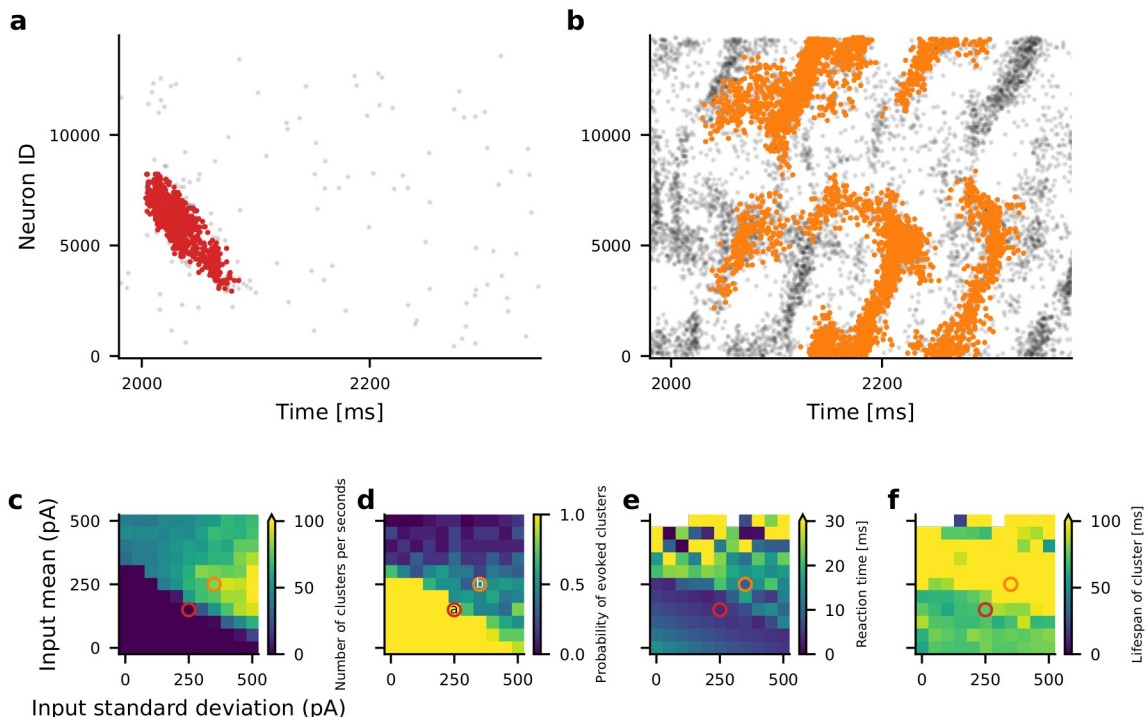

**Fig 8. Stimulus evoked STAS. (a)** Evoked STATS (red dots) in the weak input regime as a function of time. The gray dots show the background activity. **(b)** Evoked STAS in strong input regime. Note that every 5th spike is displayed. The values for input mean and input standard deviation used for panels panels **a** (red) and **b** (orange) are marked by circles of corresponding colors in panels **c-f**, respectively. **(c)** Number of clusters (STAS) as a function of the mean and standard deviation of the input current. This panel is the same as in S3b Fig. **(d)** Probability, **(e)** the reaction time and **(f)** the life span of evoked STAS as a function of of the mean and standard deviation of the input current.

dendritic arbors of some prominent neuron types are highly similar. However, it is not possible to infer from the available data whether neighboring neurons have a similar orientation of their axons. Experimental data suggest that neurons born together tend to share their inputs [36]. In addition, activity dependent plasticity may also lead to the formation of a few stronger synapses, possibly (but not necessarily) associated with a preferred projection direction. However, such mechanisms, even if viable, will only hardwire one specific set of STAS. In the following, we propose a more general and, more importantly, dynamic mechanism that may lead to asymmetric and spatially correlated connectivity of neurons that not only generates STAS, but may rapidly switch from one set of sequences to another.

Consider a network in which neurons have symmetric dendritic and axonal arbors (Fig 9a). Such a network would not support activity sequences (Fig 2) and stimulus evoked activity will be confined to the stimulated region of the network. In this network, the release of a neuromodulator (e.g. dopamine or acetylcholine) will create a phasic increase in the neuromodulator levels in small patches (Fig 9b, yellow blobs). Such patches naturally arise because of the non-uniform distribution of axons releasing the neuromodulator and its diffusion in the neural tissue, which presents an inhomogeneous medium. A similar patchy spatial profile of dopamine has been recently observed experimentally *in vivo* [37]. Most synapses within the regions of high neuromodulator concentration will be potentiated (schematically shown in Fig 9b, blue neurons) and, hence, create an asymmetric, spatially correlated connectivity for as long as the neuromodulator concentration remains high. That is, the effective connectivity along the

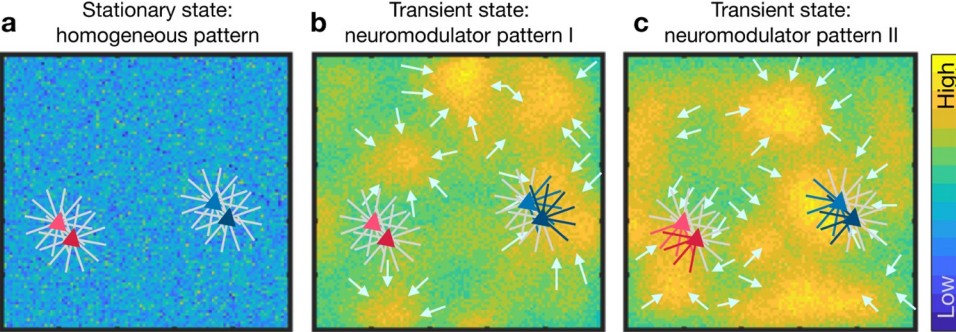

**Fig 9. Dynamic reorganization of activity sequences in a recurrent network. (a)** Schematic of a network in which neurons connect in all direction equally. The blue background shows the baseline level of a certain neuromodulator substance. Two pairs of neurons (blue and red triangles) are shown, the axons of which project in all directions uniformly. This is equivalent to the symmetric configuration and, hence, no sequential activity will emerge. **(b)** Non-uniform distribution of concentration of the neuromodulator in different parts of the network, as indicated by the colormap. The colored lines indicate the enhanced synaptic strength in specific directions. Asymmetric connectivity of neighboring neurons caused by such non-uniform neuromodulator concentration distribution may result in activity sequence in some regions of the network (e.g. neurons marked in blue). The short arrows mark the potential flow of a neuronal activity sequence along the spatial gradient of the neuromodulator concentration. **(c)** Same as in **b** for a different pattern of neuromodulator concentration which may lead to a different flow of neuronal activity, resulting in the appearance of activity sequences in a new set of neurons (e.g. those marked in red) and a change in direction of the sequence in others (e.g. those marked in blue).

spatial gradient of the neuromodulator concentration (Fig 9b, arrows) may be modified to generate STAS.

This neuromodulator based mechanism to generate spatially correlated asymmetric, anisotropic connectivity automatically provides a mechanism to rapidly switch between sequences. A change in the spatial profile of the neuromodulator concentration will potentiate another set of synapses, possibly leading to the recruitment of new neurons into the activity sequence (Fig 9b and 9c red neurons), or to the assignment of neurons to a different sequence (Fig 9b and 9c, blue neurons). In the reasoning above, we assumed that the neuromodulator enhances synaptic strengths, but the same argument also holds when the neuromodulator suppresses synaptic strengths. Thus, neuromodulators may play a crucial role, not only in the formation of STAS, but also in rapidly switching between different sets of sequences. Moreover, by their spatial concentration distribution, neuromodulators can also control the speed of the activity sequence. This idea is consistent with experimental observations, e.g. the finding that acetylcholine is important for retinal waves [38].

## Discussion

Here we have shown that spatial inhomogeneities in network connectivity can lead to the emergence of STAS in the network. Unlike existing models of sequence generation, which require either manual wiring of neurons or supervised learning, we provide a simple generative rule that renders an LCRN with the ability to generate STAS. We showed that when (1) individual neurons project a small fraction (approximately 2-5%) of their axons in a preferred direction $\phi$ (i.e. the connectivity is asymmetric), and (2) $\phi$s of neighboring neurons are similar, whereas $\phi$s of neurons further apart are unrelated (i.e. the network is anisotropic), the network will generate STAS. That is, asymmetric but spatially correlated connectivity of neurons translates into sequential spiking activity. Note that, a spatial asymmetry of neuronal connectivity can also be achieved when a neuron makes stronger instead of more synapses in the preferred direction ($\phi$). Under this mechanism, the number of STAS and their propagation velocity are

determined by the spatial extent of the connectivity asymmetry and the spatial scale of the $\phi$-correlations (Fig 6). Beyond the generation of sequences of neuronal activity, the network created using our generative rule also produces structured activity which can be used to create motor patterns, for example using supervised learning [39].

It is already well known that when neurons make connections in a spatially asymmetric manner, the network can exhibit travelling waves, which can be considered as a kind of STAS [23]. Note that, spatially asymmetric connectivity is not essential to generate travelling waves. Other mechanisms (e.g. mean and variance of the synaptic weights or imbalance of excitation and inhibition) can lead to symmetry breaking and travelling waves in networks with spatially homogeneous connectivity in excitatory neural fields [20, 40, 41] and in networks with spiking neurons [42]. However, such travelling waves are identical across the whole networks. Moreover, such travelling waves when rendered as temporal sequences (similar to what is shown in Fig 3) result in periodic activity. Here, building on previous work on networks with spatial connectivity, we showed that introducing spatially correlations in the connection asymmetry resulted in a richer dynamics of STAS, closely resembling the experimental measurements of STAS. Thus, the key novelty of our work is in demonstrating the importance of spatial correlations in the connection asymmetries for network activity sequences or correlated in- and our-degree.

While our model provides a mechanism for the emergence of spatially and temporally organized sequences of neuronal activity, it should be noted that so far experimental data have not yet revealed a clear relationship between spatial proximity of individual neurons and their order in a neuronal activity sequence. However, neighboring neurons are more correlated than those further apart [43, 44]. Furthermore, at the level of summated activity of neurons in a small neighborhood (e.g. as measured by voltage sensitive dye imaging or in local field potentials) spatially organized waves of neuronal activity are clearly visible (see review [41] and in references therein). Therefore, it is possible that the lack of evidence for spatial organization of sequences of neuronal activity could be caused by sparse sampling of neurons by extracellular electrodes or by the poor temporal resolution of calcium imaging. In case the temporal sequences of neuronal spiking activity have no spatial structure, our model still suggests that the mechanism underlying such sequences would be correlation in the in- and out-degrees of the neurons involved (see Fig 7).

Feedforward networks are simple but powerful computing devices [14, 15] and, by definition, the existence of sequential activity requires such networks. Moreover, such feedforward networks are thought to be a possible structural substrate of the *phase sequences*, proposed by Hebb to 'neuralize this behavior' [1]. Therefore, there is a general interest in understanding how feedforward networks may emerge in otherwise randomly connected networks. To this end, a number of computational studies have investigated whether Hebbian synaptic plasticity can generate such feedforward networks. These attempts were usually successful in creating feedforward networks in small random recurrent networks [45–48], but did not scale up for large recurrent networks [49]. Specifically, learning of feedforward networks using unsupervised learning rules such as Hebbian or anti-Hebbian plasticity rules resulted in a situation in which the learned feedforward network was completely disconnected from the rest of the network, unless the plasticity rule was switched off manually [49].

In general, it is assumed that networks in the brain are 'tabula rasa' and that they have to learn to generate any structured activity patterns such as STAS for behavior. However, recently it is being recognized that neuronal networks cannot learn everything using learning algorithms or synaptic plasticity rules alone and that innate connectivity structures are not completely random and have the 'innate' ability to perform certain computations [50].

Our observations of feedforward networks emerging in an LCRN with Perlin configuration provides a much simpler generative mechanism that can create feedforward networks in large

random neuronal networks without the necessity of invoking any synaptic or structural plasticity. Thus, we provide a possible mechanism by which networks in the brain can have an innate wiring that enables them to generate STAS. It can be argued that the requirement of spatially correlated asymmetry in connectivity also means that the STAS are hardwired in the network structure—not very different from supervised learning. Indeed, such connectivity requirements are too strict. Therefore, we have provided an additional mechanism by which spatial-temporal patterns of neurmodulator release can rapidly create and dynamically modulate spatially correlated asymmetry in the connectivity (see Fig 9).

## Relationship with previous network models with spatial connectivity

Neuronal networks with spatially homogeneous and isotropic connectivity have been extensively studied using neural field equations [21, 51, 52] (see review by [53]) or by numerical simulations [17, 54]. By contrast, the dynamics of LCRNs with inhomogeneities have not received much attention. In most cases, inhomogeneities were distributed uniformly in space [17, 55, 56]. Models of orientation tuning in which neurons are wired according to their tuning preferences [57] do have some resemblance with our model. However, in such models the exact spatial distribution of the connectivity has neither been studied nor was it considered relevant. In some models inhomogeneties have been introduced using mechanisms such as synaptic depression [24] and/or spike frequency adaptation [25, 26]. In such networks, because neurons cannot continue to spike forever, because of spike frequency adaptation or because of weakening of input synapses, the activity attractor is not stable and continues to move [26]. Moreover, in these networks also the spatial distribution of inhomogeneities is not relevant for the moving bump. Thus, to the best of our knowledge, our study presents the first example of a systematic study of the effects of the spatial distribution of heterogeneity on the dynamics in locally connected random networks.

## Experimental evidence supporting our connectivity rule

Our connectivity rule requires that neurons preferentially project a fraction of their connections in a given direction $\phi$ and that $\phi$s of neighboring neurons are similar. The argument can be reversed and we can also say that neurons preferentially receive a fraction of their inputs from a direction $\theta$ and that $\theta$s of neighboring neurons are similar. This implies that axonal and/or dendritic arbors of neurons should be spatially asymmetric and that axonal/dendritic arbors of neighboring neurons project in a similar preferential direction. There is plenty of evidence that most neuron types do have spatially asymmetric dendritic and axonal arbors [35, 58]. Moreover, dendritic arbors of some prominent neuron types are highly similar. However, it is not possible to determine whether the correlation between the preferential projection (reception) directions has a spatial structure. Beyond just looking at the structure of dendritic or axonal arbors, our connectivity rule also implies that neighboring neurons should share their input and/or output neurons. Experimental data suggest that neurons born together tend to share their inputs [36]. Thus, some available experimental data provide support for our connectivity rule, but more dedicated experiments are needed to check the full validity of our model.

## Model predictions

The key prediction of our network model is that in brain regions generating STAS, neurons should have asymmetric but spatially correlated network connectivity. Such correlated asymmetry can be observed in at least two different forms: (1) asymmetric but similar axonal or dendritic arbors of neighboring neurons, and (2) neighboring neurons receiving strong

synapses from common sources and sending out strong synapses to common targets. Secondly, we predict that neuromodulators may play a crucial role in the generation and control of STAS in an otherwise isotropic network. The latter can be tested by experimentally controlling the spatial profile of the corresponding neuromodulator release pattern by optogenetic stimulation of its source neurons.

Effectively, the asymmetric but spatially correlated connectivity of neurons implies correlated in- and out-degrees of neurons. Therefore, our model predicts that even when asymmetric but similar axonal or dendritic arbors of neighboring neurons are not observed, networks with intrinsically generated sequential activity in the neurons should have correlated in- and out-degree of these neurons.

In summary, we propose a simple generative rule that enables neuronal networks to generate STAS. How these spontaneously generated sequences interact with stimuli and how we can create stimulus—sequence associations is an interesting and involved question that will be addressed in future work. Similarly, more work is needed to determine the role of neuronal and synaptic weight heterogeneities in shaping spontaneous and stimulus-evoked neuronal activity sequences, either with or without changes in neuromodulator concentration distributions.

## Methods

### Neuron model

Neurons in the recurrent networks were modelled as 'leaky-integrate-and-fire' (LIF) neurons. The sub-threshold membrane potential ($v$) dynamics of LIF neurons are given by:

$$C_m \frac{dv_i}{dt} = -g_L(v_i(t) - E_L) + I_i(t) + \mu_{GWN_i} + \sigma_{GWN_i} \tag{1}$$

where $\tau_m = \frac{C_m}{g_L}$ denotes the membrane time constant, $C_m$ the membrane capacitance, $g_L$ the leak conductance, $E_L$ the leak reversal potential, $I(t)$ the total synaptic current and $\mu_{GWN_i}$ and $\sigma_{GWN_i}$ are the mean and standard deviation of Gaussian white noise input to the neuron. The neuron parameters are listed in Table 1.

**Table 1. Parameter values for the neurons (top) and for the synapses (bottom) in both network models.**

| Neurons & synapses | | |
|---|---|---|
| **Name** | **Value** | **Description** |
| $C_m$ | 250.0 pF | Membrane capacitance |
| $g_L$ | 25.0 nS | Membrane capacitance |
| $\tau_m$ | 10.0 ms | Membrane time constant |
| $E_L$ | - 70.0 mV | Leak potential, resting potential |
| $V_{th}$ | - 55.0 mV | Spike threshold |
| $V_{reset}$ | - 70.0 mV | Resting membrane potential |
| $t_{ref}$ | 2.0 ms | Refractory period |
| $\tau_{exc}$ | 5.0 ms | Time constant of excitatory synapse |
| $\tau_{inh}$ | 5.0 ms | Time constant of inhibitory synapse |
| $J_{ext}$ | 1.0 pA | Synaptic weight of the external input |
| $J_x$ | 10.0 pA | Base value of the synaptic weight |
| $J_{exc}$ | 0.22 mV | Amplitude of excitatory post synaptic potential |
| $J_{inh}$ | 0.22 mV | Amplitude of inhibitory post synaptic potential |
| $d$ | 1.0 ms | Synaptic delay |

## Synapse model

Synapses in the network were modelled as current transients. The temporal profile of the current transient in response to a single pre-synaptic spike was modelled as an $\alpha$ function:

$$I_{syn} = J_{syn} \frac{t - t_{spk}}{\tau_{syn}} \exp\left( -\frac{t - t_{spk}}{\tau_{syn}} \right) \qquad (2)$$

where $J_{syn}$, $\tau_{syn}$ and $t_{spk}$ denote synaptic amplitude, synaptic time constant and spike time, respectively. The term *syn* stands for *exc* and *inh* for excitatory and inhibitory synapses, respectively.

We adjusted the synaptic currents to obtain weak synapses, such that both a unitary inhibitory post-synaptic potential ($J_{inh}$) and a unitary excitatory postsynaptic potential ($J_{exc}$) had an amplitude of 0.22 mV. The synapse parameters (synaptic strength, time constant and delay) were fixed throughout the simulations and are listed in Table 1, bottom.

## Network architecture

We studied two types of recurrent network models in which the connection probability between any two neurons depended on the physical distance between them. Neurons in both network models were placed on a regular square grid. To avoid boundary effect, the grid was folded to form a torus [17]. In both network types, multiple connections were permitted (S1a Fig), but self-connections were excluded.

*Networks with only inhibitory neurons*: The first type network model (I-network) was composed of only inhibitory neurons. The brain regions striatum and central amygdala are examples of purely inhibitory recurrent networks in the brain. These neurons were arranged on a 100×100 grid (*npop* = 10, 000). Each neuron projected to 1,000 other neurons (corresponding to an average connection probability in the network of 10%). The distance-dependent connection probability varied according to a $\Gamma$ distribution [30] with the following parameters: $\kappa = 4$ for the shape and $\theta = 3$ for the spatial scale. All I-network parameters are summarized in Table 2.

*Networks with both excitatory and inhibitory neurons*: The second type network model (EI-network) was composed of both excitatory and inhibitory neurons. EI-networks are more common in the brain than I-networks, and are observed throughout the neocortex and CA3

**Table 2. Parameter values for the networks (top), for the connections (middle) and for an external input (bottom) in I network model.**

| I network model | | |
|---|---|---|
| **Name** | **Value** | **Description** |
| Neuron model | | Integrate and fire |
| *nrow, ncol* | 100 | number of rows/columns in a network layer |
| *npop* | *nrow* * *ncol* = 10,000 | number of neurons |
| Synapse model | | $\alpha$ function, current-based model |
| $\kappa$ | 4 | Shape for gamma distribution function |
| $\theta$ | 3 | Scale for gamma distribution function |
| $p_{conn}$ | 0.1 | Average connection probability of target neurons |
| $n_{conn}$ | $p_{conn}$ * *npop* = 1000 | Number of recurrent connections per neuron |
| $J_{rec}$ | $-J_x$ = -0.22 mV | Synaptic weight of recurrent inhibitory connections |
| $\mu_{GWN}$ | 700.0 pA | Mean of external GWN input |
| $\sigma_{GWN}$ | 100.0 pA | Standard deviation of external GWN input |

**Table 3. Parameter values for the networks (top), for the connections (middle) and for an external input (bottom) in EI-network model.**

| EI network model | | |
|---|---|---|
| **Name** | **Value** | **Description** |
| Neuron model | | Integrate and fire |
| $nrow_E$, $ncol_E$ | 120 | number of rows/columns in exc. network layer |
| $nrow_I$, $ncol_I$ | 60 | number of rows/columns in inh. network layer |
| $npop_E$ | $ncol_E \times nrow_E = 14{,}400$ | number of excitatory neurons |
| $npop_I$ | $ncol_I \times nrow_I = 3{,}600$ | number of inhibitory neurons |
| $npop_{ratio}$ | $npop_E : npop_I = 4 : 1$ | Ratio of exc.—inh. neurons |
| Synapse model | | $\alpha$ function, current-based model |
| $\sigma_{EE}$ | 9 | Space constant for E→E connectivity |
| $\sigma_{IE}$ | 4.5 | Space constant for E→I connectivity |
| $\sigma_{EI}$ | 12 | Space constant for I→E connectivity |
| $\sigma_{II}$ | 6 | Space constant for I→I connectivity |
| $p_{conn}$ | 0.05 | Connection probability of target neurons |
| $n_{connE}$ | $p_{conn} \times npop_E = 720$ | Connection number of excitatory targets |
| $n_{connI}$ | $p_{conn} \times npop_I = 180$ | Connection number of inhibitory targets |
| $g$ | 8 | Ratio of recurrent inhibition and excitation |
| $J_E$ | $J_x = 0.22$ mV | Synaptic weights of excitatory source |
| $J_I$ | $g \times J_E = -1.76$ mV | Synaptic weights of inhibitory source |
| $\mu_{GWN}$ | 350.0 pA | Mean of external GWN input |
| $\sigma_{GWN}$ | 100.0 pA | Standard deviation of external GWN input |

region of the hippocampus. Excitatory and inhibitory neurons were arranged on a 120×120 ($npop_E = 14, 400$) and on a 60×60 grid ($npop_I = 3, 600$), respectively. Each neuron of the excitatory and inhibitory populations projected to 720 excitatory and 180 inhibitory neurons (average connection probability 5%). The connection probability varied with distance between neurons according to a Gaussian distribution [17, 30, 59]. The space constant (standard deviation of the Gaussian distribution) for excitatory source projected to excitatory targets was $\sigma_{EE}$ = 9 ($\sigma_{EI}$ = 4.5 for inhibitory source) and to inhibitory targets $\sigma_{IE}$ = 12 ($\sigma_{II}$ = 6 for inhibitory source). We considered a high probability of connections within a small neighborhood, therefore, these networks were referred to as locally connected random networks (LCRN [54]). All EI-network parameters are summarized in Table 3. Whenever possible, we used parameters corresponding to a standard EI-network [60].

## Asymmetry in spatial connections

Typically, in network models with distance-dependent connectivity, the connection probability is considered to be isotropic in all directions. In the network models studied here, however, we deviated from this assumption and introduced spatial inhomogeneities in the recurrent connections. Specifically, we considered a scenario in which the neuronal connectivity was asymmetric in the sense that each neuron projected a small fraction of its axons in a particular direction $\phi$ (Fig 1a and 1b). At the same time we ensured that the out-degree of each neuron was the same as in an LCRN with isotropic connectivity. In I-networks all the neurons had asymmetric connectivity, whereas in EI networks only connection asymmetry was introduced only for excitatory to excitatory projections. The fraction of extra connections in the direction $\phi$ depended on the shift in the region of post-synaptic neurons (green circle in Fig 1a). To quantify the change in connection probability, we estimated the average distance-dependent

connectivity in the symmetric configuration ($S$) and in the homogeneous configuration ($H$). The change in connectivity was then measured as $(S − H)/S$.

Shifting the connectivity region of a neuron by one grid location in our network model means that the probability of a neuron to make a connection in that direction was increased or decreased by some amount $\Delta p$ § ($0 − 100\%$), depending on the distance between the neurons. At short distances, the connection probability almost doubled, whereas at distances between 10-20 grid points, there was only a very small change in the connection probability (S1b Fig). At larger distances ($> 20$ grid points), the connection probability changed by a large amount (S1b Fig). This is because at such distances the connectivity is sparse (connection probability $< 0.01$) and the measure $(S − H)/S$ amplifies small changes for small $S$. Because we maintained the out-degree of the neurons, an increase in the connection probability in one direction implied a reduction in connection probability by the same amount in the opposite direction.

Note that an increased connection probability also increased the probability to form multiple connections in the close neighborhood of the projecting neuron (S1a Fig). The preferred direction ($\phi$) for each neuron was chosen at random from a set of eight different directions, considering that neurons were positioned in a grid pattern. All other synaptic parameters, such as the number of total connections, the space constant of the connectivity kernel and the synaptic weights were identical for all neurons.

## Spatial distribution of asymmetry in spatial connections

In a network model with asymmetric recurrent connections it does not suffice to select the preferred connectivity direction of target neurons for individual source neurons depending on their positions. We also need to define how exactly the 'directions' ($\phi$) are distributed in space. For this, we considered four qualitatively different configurations.

*Homogeneous configuration*: In this configuration all neurons had the same $\phi$, indicating a single-direction bias of the projections of all neurons (Fig 1c, left).

*Random configuration*: In this configuration $\phi$ for each neuron was chosen independently at random from a uniform distribution (Fig 1c, middle).

*Perlin configuration*: In this configuration $\phi$ was also chosen from a uniform distribution as in the random configuration, but it was assigned to each neuron according to a 2-D Perlin noise—a class of gradient noise [61]. The generation of Perlin noise, which amounts to a simple procedure to introduce spatial correlations in an otherwise random distribution, is described below. This spatial distribution of $\phi$ ensured that neighboring neurons had similar preferred directions (Fig 1c, right).

*Symmetric configuration*: In this configuration all neurons established connections in an isotropic manner, without any directional preference.

## Perlin noise

To generate Perlin noise we first created a p×p grid (Perlin grid) that covered the whole network (of size N×N; $N = 100$ I-networks and $N = 120$ for EI-networks). We defined $p = \frac{N}{Perlin\ scale}$. For example a *Perlin scale* = 20 meant that the Perlin grid was of size 5×5 for I-networks and 6×6 for EI-networks. The variable *Perlin scale* controlled the spatial scale of the correlations. After defining the Perlin grid, each grid point was assigned a value chosen from a uniform distribution $\mathbb{U}[0, 2\pi]$. Next, we interpolated the Perlin grid to a size of the N×N (same size as the I-network or EI-network). The resulting value was used as the $\phi$ of the neuron located at that grid point. For more details about the generation of Perlin noise please see [61].

## Input and network dynamics

All neurons received independent, homogeneous excitatory inputs from an external drive. We selected Gaussian white noise as an adequate input for generating ongoing spiking activity dynamics, which could be set to different activity levels by varying the input mean and variance independently.

## Spectral analysis

To characterize oscillations in the network activity, we estimated the spectrum of neuronal activity. To this end we summed the activity of a population of neurons. This summed network activity was obtained by three different procedures: by summing the activity of all neurons, by summing the activity of the neurons from a 10×10 region in the network, and by summing the activity of 100 randomly chosen neurons from the entire network. In each case, the neuronal activity was binned using 5 ms wide rectangular bins (i.e. sampling frequency = 200 Hz). The power spectrum was obtained by using the function `scipy.signal.pwelch` function from the `signal` toolbox of the `SciPy` python package. We estimated the spectrum by setting nfft = 4096. For the estimation of the spectrogram (S3 Fig) we estimated the spectrum of the network activity for shorts epochs (epoch length = 200 ms; overlap = 50 ms).

## Identification of spatio-temporally clustered activity

To identify the STAS we rendered the spiking activity in a three-dimensional space spanned by two spatial dimensions of the network and one time dimension. Each spike is a point and a STAS is a cluster in this 3-D space. We used the density-based spatial clustering algorithm of applications with noise (DBSCAN) [62] to determine individual clusters of the spiking activity in space and time. The DBSCAN algorithm required two parameters for the analysis: the maximum distance between two points in a cluster (*eps*) and the minimum number of points required to form a cluster (*minPts*). This algorithm needed a supervised control and an adequate value for *eps*, depending on the average spatial and temporal distance (i.e. inter-spike interval) between spikes of the neurons. For instance, when the average firing rate of the neurons was too high, then multiple STAS could be coalesced into a single STAS. On the other hand, when the average firing rate of the neurons was small, a single STAS could be dissociated into multiple small STAS. To avoid such problems, we reduced the temporal scale of spikes by a factor 20 and 3-5 for I-networks and EI-networks, respectively. Note that this temporal rescaling was not used for the estimation of other properties of the network activity dynamics. The *eps* value was set to 3 and 3-4 for I-networks and EI-networks, respectively. Using the DBSCAN we identified STAS in successive, overlapping time windows of duration 1 sec (overlap duration 0.9 sec).

## Spatial arrangement of locally clustered activity

For each identified cluster, we calculated the spatial centroids of activity bumps observed in successive time windows of 1 sec. The vectors composed of these successive centroids described the successive spatial coordinates of the bump activity and, hence, revealed the movement of the bump activity. Using these vectors we plotted the pathways of the moving activity bumps in the network's spatial map.

## Quantification of activity bump movement

Keeping track of direction changes in bump movement is an adequate measure for the dynamics of sequential and non-sequential activities. In travelling waves, activity bumps typically

move in a single direction, whereas in spatially fixed patterns, activity bumps alternate directions erratically over very short time scales. Between these two extrema, activity bump movements changed their direction slowly. The directional change of bump movement is given by:

$$d\alpha = \alpha_t - \alpha_{t-1} \tag{3}$$

where $\alpha_t$ denotes the direction of bump movement observed at time step $t$, and $d\alpha$ ranges from $-\pi$ (opposite direction) over 0 (no alternation) to $\pi$ (opposite direction).

### Identification of feedforward networks in the LCRN

To identify a feedforward network *FF* in the LCRN we started with a set of 64 neurons ($F_i$), located in an 8×8 region in the network. This choice was motivated by our observation that individual spatial clusters of active neurons were typically of size 8×8. We then identified all post-synaptic neurons ($P_i$) connected to any of the neurons in the set $F_i$. From the set $P_i$ we selected the 64 neurons ($F_{i+1}$) that received the most number of connections from the $F_i$. We repeated this procedure 50 times, starting at 100 different, randomly selected locations. Given the delays in the network, 50 time steps would imply that a sequence lasted for at least 100 ms. In this manner we identified feedforward networks with excitatory (inhibitory) connections from $F_n$ to $F_{n+1}$ in EI-networks (I-networks).

To quantify the feedforward path we measured the number of neurons (*nFF*) belonging to $F_1 \ldots F_{50}$ over the trajectory between the centroids of $F_1$ and $F_{50}$ (Fig 7b and 7c). Note that each neuron was counted only once. The larger *nFF*, the longer and/or broader was the feedforward network.

In addition, we measured the *Effective length* as the Euclidean distance between the centroids of $F_1$ and $F_{50}$. Based on visual inspection of the locations of $F_1 \ldots F_{50}$, we checked that $\{F_n; n > 2\}$ did not loop back to the same region where $F_1$ was located.

To call the set of neurons that constitute $F_1 \ldots F_{50}$ a feedforward path, capable of creating STAS, we argued that $\{F_n; n > 2\}$ must be outside the connection region of the neurons in $F_1$. In EI-networks, the space constant of excitatory projections of a neurons was $\sigma_E = 12$. If we assume that ≈ 70% connections are within one $\sigma_E$ (because the shape of connection probability function is Gaussian), then in EI-networks the combined connection region of all neurons in $F_1$ has a diameter of 12 + 8 + 12 = 32. Therefore, to be outside the connection region of $F_1$, the centroid of $F_{50}$ should be at least 16 grid points away from the centroid of $F_1$ in EI-networks, that is, the *Effective length* should be larger that 16. Similarly, we estimated the *Effective length* for I-networks as 16.

Thus, we defined that an effective feedforward pathway capable to creating STAS should have an *Effective length* > 16 (for both EI- and I-networks). Finally, we defined *pFF* as the frequency of finding a feedforward path of *Effective length* > 16.

### Evoked STAS

To study stimulus evoked STAS we stimulated ≈ 50 excitatory neurons in the EI-network model by injecting direct current with an amplitude of 500 pA. This input was in addition to the background input every neuron received. The stimulus input lasted for 50 ms. To collect sufficient data for further statistical analysis, each stimulus was presented 20 times to the stimulated neurons. We measured the probability of evoking a STAS, the network reaction time and the lifespan of evoked STAS. The probability of evoked STAS was calculated as $C_e/N_{stim}$, where $C_e$ is the count of evoked STAS and $N_{stim} = 20$ is the number of times the neurons were stimulated. An evoked STAS was characterized by the following criteria: the existence of clustered spikes in the stimulated neurons within 50 ms of stimulus onset and the absence of

clustered spikes in the stimulated neurons for 50 ms prior to stimulus onset. The reaction time was defined as the time the stimulated neurons took to elicit their first spikes after stimulus onset. The lifespan was defined by the time difference of the first and the last spike of the evoked STAS.

## Simulation tools

All simulations of the network models were performed using the NEST simulation software (http://www.nest-initiative.org) [63]. The dynamical equations were integrated at a fixed temporal resolution of 0.1 ms. Simulation data were analyzed with Python using the scientific libraries SciPy (http://www.scipy.org) and NumPy (http://www.numpy.org), and visualized using the plotting library Matplotlib (http://matplotlib.org). The code to simulate the model is available at GitHub https://github.com/babsey/spatio-temporal-activity-sequence.

## Supporting information

**S1 Fig. Multiplicity of connections and effect of forcing a neuron to make some connections preferentially in the direction $\phi$. (a)** Count distribution of multiple connections between any pair of neurons in an I-network (left) and in an EI-network (right). The multiple connections were formed primarily because of the connectivity rule (local connectivity). Note that the network configuration (as indicated by different colors of the curves) had only a minute influence on the distribution of multiple connections. **(b:left)** Average $\delta_{conn}$ for I-networks (average over all the neurons in the network). **(b:right)** Same as in the left panel, but for EI-networks. Forcing a neuron to make preferentially connections in the direction $\phi$ increased its connectivity in that direction: connectivity doubled in the immediate vicinity. Correspondingly, the connectivity was reduced by the same amount in the opposite direction. This change in the opposite direction is because we achieved asymmetry by shifting the connectivity cloud in the direction specified by $\phi$ (Fig 1). That is, in the immediate vicinity, the connection probability was doubled in the direction $\phi$. This increase may look very large, but it nevertheless was not large enough to alter the probability of multiple connections in the network (a). Note that there is connectivity increase and corresponding decrease at larger distances, but such change was not of much consequence because at these large distances the connection probability was very small to begin with.
(TIFF)

**S2 Fig. Effect of input and excitation-inhibition balance on the emergence of spatio-temporal sequences in an EI-network. (a)** Average firing rate of excitatory neurons as a function of the mean (ordinate) and standard deviation (abscissa) of the input noise to the all neurons. **(b)** The probability of observing an evoked STAS (cluster) as a function of the mean (ordinate) and standard deviation (abscissa) of the input noise to the all neurons. **(c)** The life span of an evoked STAS as a function of the mean (ordinate) and standard deviation (abscissa) of the input noise to the all neurons. The values for the excitation-inhibition balance used for panels **a-c** are marked by the orange circle in panels **d-f**. **(d)** Average firing rate of excitatory neurons as a function of excitatory synaptic weight (ordinate) and the ratio of recurrent inhibition and excitation ($g$, abscissa). **(b)** The probability of observing an evoked STAS as a function of excitatory synaptic weight (ordinate) and $g$ (abscissa). **(c)** The life span of an evoked STAS as a function of excitatory synaptic weight (ordinate) and $g$ (abscissa). The values of input mean and standard deviation used for panels **d-f** are marked by the orange circle in panels **a-c**.
(TIFF)

**S3 Fig. Spectrogram of the population activity in EI-network connected according to the Perlin configuration. (a)** Spectrogram for the EI-network with weaker recurrent synaptic strength ($Ji = 10pA$). The spectrogram was estimated by splitting the time series of population activity (bin width 5 ms) in 200 ms windows. Consecutive epochs overlapped for 150 ms duration. **(c)** Power in individual epochs (black curves) and the mean power (red curve) corresponding to the spectrogram shown in panel **a** as a function of frequency. **(b)** Same as in panel **a** for stronger recurrent inhibitory synaptic strength ($Ji = 20\ pA$). **(d)** Same as in panel **c** but for the spectrogram shown in panel **b**.
(TIFF)

**S1 Video. Spiking activity in I-networks for the four different configurations.** Each panel shows the activity of inhibitory neurons observed over a time window of 50 ms (disjoint windows), rendered in the two-dimensional network space. Each dot represent a spike of the neuron located at that grid point. Active neurons are colored to identify individual spatio-temporal activity sequences (STAS). Spikes rendered in the same color belong to the same STAS and in yellow color are not part of an STAS.
(MP4)

## Acknowledgments

We would like to thanks Dr. Upinder Singh Bhalla, Dr. Ulrich Egert, Dr. Thilo Womelsdorf, Dr. Jyotika Bahuguna and Ramon H. Martinez for helpful discussions during the preparation of the manuscript.

## Author Contributions

**Conceptualization:** Arvind Kumar.

**Data curation:** Arvind Kumar.

**Formal analysis:** Sebastian Spreizer, Arvind Kumar.

**Funding acquisition:** Ad Aertsen, Arvind Kumar.

**Investigation:** Sebastian Spreizer, Arvind Kumar.

**Methodology:** Sebastian Spreizer, Arvind Kumar.

**Project administration:** Arvind Kumar.

**Resources:** Arvind Kumar.

**Software:** Arvind Kumar.

**Supervision:** Ad Aertsen, Arvind Kumar.

**Validation:** Arvind Kumar.

**Visualization:** Sebastian Spreizer, Arvind Kumar.

**Writing – original draft:** Arvind Kumar.

**Writing – review & editing:** Sebastian Spreizer, Ad Aertsen, Arvind Kumar.

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
