## [Decision Letter · Decision Letter 0]

30 Aug 2019

Dear Dr Kumar,

Thank you very much for submitting your manuscript, 'From space to time: Spatial inhomogeneities lead to the emergence of spatiotemporal sequences in spiking neuronal networks', to PLOS Computational Biology.

First of all let me apologize for the delay in receiving this decision. We can assure that all efforts were done to speed up the process, but it simply fell in the long tail of the distribution, exasperated by the summer break.

As with all papers submitted to the journal, yours was fully evaluated by the PLOS Computational Biology editorial team, and in this case, by independent peer reviewers. The reviewers appreciated the attention to an important topic but identified some aspects of the manuscript that should be improved.

Reviewer 1 reaised some important points. While as Editors we think that even an abstract model can provide insight on a biological system, and are thus considering this paper within the scope of the journal, we strongly suggest to stress the relevance of your findings, and make the methods more explicit and reproducible.

We would therefore like to ask you to modify the manuscript according to the review recommendations before we can consider your manuscript for acceptance. Your revisions should address the specific points made by each reviewer and we encourage you to respond to particular issues Please note while forming your response, if your article is accepted, you may have the opportunity to make the peer review history publicly available. The record will include editor decision letters (with reviews) and your responses to reviewer comments. If eligible, we will contact you to opt in or out.raised.

- Supporting Information uploaded as separate files, titled 'Dataset', 'Figure', 'Table', 'Text', 'Protocol', 'Audio', or 'Video'.

We hope to receive your revised manuscript within the next 30 days. If you anticipate any delay in its return, we ask that you let us know the expected resubmission date by email at ploscompbiol@plos.org.

Sincerely,

Daniele Marinazzo

Deputy Editor

PLOS Computational Biology

Daniele Marinazzo

Deputy Editor

PLOS Computational Biology

[LINK]

Reviewer's Responses to Questions

**Comments to the Authors:**

Reviewer #1: The authors consider the question of how recurrent spiking neural network models can generate spatiotemporal sequences of activity, such as those believed to underly some neural comptuations. Through simulations and some intuition from the eigenvalue spectrum of their networks, they find that these sequences tend to be reliably generated when network connectivity satisfies a few reasonable and intuitive conditions. Notably, they included locally correlated anisotropies in the connectivity structure, which has not been considered in previous work to my knowledge, and which seems to be a key to generating coherent spatiotemporal sequences.

They additionally looked at the interaction between stimuli and these activity sequences, and proposed a mechanism through which neuromodulators might endow a network with the connectivity properties necessary for their generation. The study is interesting, novel, and potentially important for understanding dynamical computations in biologically plausible spiking networks.

The manuscript is suitable for publication in PLoS Comp Bio without any further changes.

Reviewer #2: This manuscript develops a model of the potential mechanisms underlying the spatiotemporal patterns of neural activity observed in vivo. The focus is on neural sequences in the form of a sequential activation of neurons (spatiotemporal activity sequences-STAS). The authors propose that these patterns emerge as a result of shared asymmetries in the direction of axonal projections within cortical networks, specifically, neighbouring neurons may have a bias to project in a random but shared direction. Thus creating preferred channels by which activity will flow. The results seem solid, and to strongly support the conclusions, furthermore the model seems fairly systematic. The paper certainly adds to our understanding of the potential influence of shared axonal projection anisotropies in cortical networks, my main concern however, in the context of the current journal, revolves around biological relevance.

The mechanisms underlying sequence generation is an important one, and the authors cite many experimental papers that have observed these patterns (ref. 3-11). A potential concern is that to the best of my knowledge none of these papers suggest that there is any spatial organization to these observed patterns, indeed as I understand it in most cases it is explicitly not the case that neurons that fire closer together in time are located close together in space—this certainly does not seem to be the case in the birdsong system as I believe recent work by Michael Long suggests. It is stated that “the key prediction” (li 474) of the models is that brain regions generating STAS, should have asymmetric network connectivity. It seems that first and foremost that the key prediction/assumption is that STASs are spatially organized, and while I may be mistaken, to the best of my knowledge this assumption is not consistent with the data.

Figure 4. It is stated that both I and EI networks exhibited clear oscillations in all configurations. I was not able to clearly see these oscillations in the power spectra of Figure 4, particularly for the EI random networks. In many cases the peaks were not particularly obvious, or in the range of 30-60 Hz, or was it clear that they were significant.

Since the goal of the paper is to present a biologically plausible model I’m not sure the I-networks really contributed to the paper as I-networks are not biologically entities.

It would be helpful to provide the equations for the synaptic currents.

**Have all data underlying the figures and results presented in the manuscript been provided?**

Reviewer #1: No: The authors stated that code will be made available on github prior to publication.

Reviewer #2: Yes

PLOS authors have the option to publish the peer review history of their article (what does this mean?). If published, this will include your full peer review and any attached files.

Reviewer #1: No

Reviewer #2: No

---

## [Editor Report · Decision Letter 1]

24 Sep 2019

Dear Dr Kumar,

We are pleased to inform you that your manuscript 'From space to time: Spatial inhomogeneities lead to the emergence of spatiotemporal sequences in spiking neuronal networks' has been provisionally accepted for publication in PLOS Computational Biology.

Before your manuscript can be formally accepted you will need to complete some formatting changes, which you will receive in a follow up email.

Please be aware that it may take several days for you to receive this email; during this time no action is required by you. Once you have received these formatting requests, please note that your manuscript will not be scheduled for publication until you have made the required changes.

In the meantime, please log into Editorial Manager at https://www.editorialmanager.com/pcompbiol/, click the "Update My Information" link at the top of the page, and update your user information to ensure an efficient production and billing process.

One of the goals of PLOS is to make science accessible to educators and the public. PLOS staff issue occasional press releases and make early versions of PLOS Computational Biology articles available to science writers and journalists. PLOS staff also collaborate with Communication and Public Information Offices and would be happy to work with the relevant people at your institution or funding agency. If your institution or funding agency is interested in promoting your findings, please ask them to coordinate their releases with PLOS (contact ploscompbiol@plos.org).

Thank you again for supporting Open Access publishing. We look forward to publishing your paper in PLOS Computational Biology.

Sincerely,

Daniele Marinazzo

Deputy Editor

PLOS Computational Biology

Daniele Marinazzo

Deputy Editor

PLOS Computational Biology

---

## [Editor Report · Acceptance letter]

18 Oct 2019

PCOMPBIOL-D-19-00990R1 

From space to time: Spatial inhomogeneities lead to the emergence of spatiotemporal sequences in spiking neuronal networks

Dear Dr Kumar,

I am pleased to inform you that your manuscript has been formally accepted for publication in PLOS Computational Biology. Your manuscript is now with our production department and you will be notified of the publication date in due course.

With kind regards,

Laura Mallard
